# S2J: Bridging the Gap Between Solving and Judging Ability in Generative Reward Models

## Abstract

With the rapid development of large language models (LLMs), generative reward models (GRMs) have been widely adopted for reward modeling and evaluation. Previous studies have primarily focused on training specialized GRMs by optimizing them on preference datasets with the judgment correctness as supervision. While it's widely accepted that GRMs with stronger problem-solving capabilities typically exhibit superior judgment abilities, we first identify a significant solve-to-judge gap when examining individual queries. Specifically, the solve-to-judge gap refers to the phenomenon where GRMs struggle to make correct judgments on some queries (14%-37%), despite being fully capable of solving them. In this paper, we propose the Solve-to-Judge (S2J) approach to address this problem. Specifically, S2J simultaneously leverages both the solving and judging capabilities on a single GRM's output for supervision, explicitly linking the GRM's problem-solving and evaluation abilities during model optimization, thereby narrowing the gap. Our comprehensive experiments demonstrate that S2J effectively reduces the solve-to-judge gap by 16.2%, thereby enhancing the model's judgment performance by 5.8%. Notably, S2J achieves state-of-the-art (SOTA) performance among GRMs built on the same base model while utilizing a significantly smaller training dataset. Moreover, S2J accomplishes this through self-evolution without relying on more powerful external models for distillation.

## 1 Introduction

As Large Language Models (LLMs) continue to evolve rapidly, a variety of evaluation paradigms have been proposed to accurately evaluate the quality of their responses. This is not only crucial for providing accurate reward signals in post-training (Ouyang et al., 2022; Bai et al., 2022; Wang et al., 2024a), but also important for automated evaluation and benchmark construction (Zheng et al., 2023; Dubois et al., 2024). Among them, Generative Reward Mmodels (GRMs) have been proposed as a solution, which treats evaluation as a capability of LLMs and leverages LLMs to evaluate other LLMs (Zheng et al., 2023; Li et al., 2025). Unlike scalar reward models, which only output a single numerical score (Liu et al., 2024a; Lambert et al., 2024), GRMs utilize the generative capabilities of LLMs to produce an interpretable analysis before rendering a verdict. Due to the extra analysis process, GRMs often lead to more accurate judgments (Gu et al., 2024; Li et al., 2025).

LLM-as-a-Judge represents the first proposed generative reward model method, which is simply implemented through basic prompt engineering technique (Zheng et al., 2023; Saha et al., 2023; Zhong et al., 2025). Subsequently, researchers employ methods such as Supervised Fine-Tuning (SFT) and Direct Preference Optimization (DPO) (Rafailov et al., 2023) to train specialized models on judging tasks (Yu et al., 2025a; Wang et al., 2024c; Ye et al., 2024b; Wang et al., 2024b; Zhang et al., 2025), thereby enhancing their judging capabilities. More recently, some studies have treated generative reward modeling as a reasoning task, aiming to incentivizes deep thinking abilities in judgment tasks of GRMs through Reinforcement Learning with Verifiable Rewards (RLVR) (Chen et al., 2025; Guo et al., 2025b; Whitehouse et al., 2025; Huang et al., 2025). The correctness of judgment naturally serves as the reward signal for optimization in this process.

A common consensus in the community suggests that LLMs with stronger problem-solving abilities also possess stronger judgment capabilities (Lambert et al., 2024; Malik et al., 2025; Liu et al., 2024c; Tan et al., 2024). However, by examining individual queries, we observe an interesting

phenomenon: sometimes an LLM can correctly solve a query, yet when tasked with evaluating a response to that same query, it fails to successfully analyze the query and then produce wrong judgment. To further examine the phenomenon, we then conduct a broader investigation into this phenomenon and draw two key findings: (1) On average, LLMs with superior problem-solving abilities indeed exhibit better judgment capabilities, aligning with previous research (Lambert et al., 2024). (2) At the individual query level, we discover that models are not consistently able to correctly evaluate queries, even when they are fully capable of solving that same query, revealing what we term **solve-to-judge gap**. Further investigation indicates that this gap emerges from a degradation in solving performance when models shifts from solving scenarios to evaluation scenarios.

In this paper, we introduce **Solve-to-Judge (S2J)** to address this problem by incorporating rewards for problem-solving during the judging process, rather than focusing solely on judging correctness for optimization. For judging correctness, we follow previous work (Guo et al., 2025b; Chen et al., 2025) to provide a higher reward if the model produces the correct judgment result. For problem-solving capability during the judging process, we require the model to first solve the user's query itself before executing the judgment. We consider two scenarios: if the query has a ground truth (like mathematics problems with numerical solutions), we use rule-based methods to verify the correctness of the self-generated solution. If the query is a subjective task without a ground truth (such as creative writing), we employ a separate scalar reward model as a scorer to assign the reward. To ensure accuracy, we only apply the scalar reward model when its judgment result is correct.

We evaluate our approach on four reward model benchmarks, including PPE Correctness (Frick et al., 2024), PPE Preference (Frick et al., 2024), Reward Bench (Lambert et al., 2024), and RMB (Zhou et al., 2024). Our S2J achieves substantial improvements over the base model, with average judgment accuracy increasing from 67.0% to 72.7%. Moreover, our method outperforms the current SOTA open-source model built on the same model series while using significantly less training data. More importantly, we use the probability of incorrect judgment given correct solution $P(j = 0 \mid s = 1)$ to measure the solve-to-judge gap. Our model reduces this gap by 16.2% compared to the base model, showing a 9.6% improvement over SOTA which reduce it by 6.6%, and a 9.3% improvement over using only correct judgment results as rewards which reduced it by 6.9%. Our in-depth analyses show that S2J indeed effectively enable models to leverage their intrinsic problem-solving knowledge for judging tasks, providing evidence for our claims.

Our contributions can be summarized as:

- We first reveal and identify the solve-to-judge gap problem through extensive experiments. We point out that current GRMs fail to correctly evaluate 14%-37% of problems they can accurately solve, establishing the comprehensive understanding of the solve-to-judge gap limitation.

- We introduce S2J, an approach that narrows the solve-to-judge gap by jointly optimizing for judging capability and solving capability, effectively reducing the solve-to-judge gap and thereby improving the judgment capability of GRMs.

- We demonstrate S2J's significant reduction in the solve-to-judge gap (-16.2%) and improvement in model's judgment performance (+5.8%) across multiple benchmarks.

## 2 THE SOLVE-TO-JUDGE GAP: AN EMPIRICAL ANALYSIS

### 2.1 TASK DEFINITION

In this paper, we focus on the pairwise judging task, which evaluates a model's ability to compare and select the better response between two candidates. Formally, we consider a preference dataset

$$\mathcal{D} = \{(x^{(i)}, y_a^{(i)}, y_b^{(i)}, l^{(i)})\}_{i=1}^N, \tag{1}$$

where $x$ is an input prompt, $(y_a, y_b)$ is a pair of candidate responses, and $l \in \{a, b\}$ is a preference label indicating which response is preferred.

### 2.2 REINFORCEMENT LEARNING WITH VERIFIABLE OUTCOME REWARD

Recent studies (e.g., RM-R1 (Chen et al., 2025), RRM (Guo et al., 2025b)) have preliminarily explored using the outcome reward as the reward signal for RLVR to optimize a generative reward

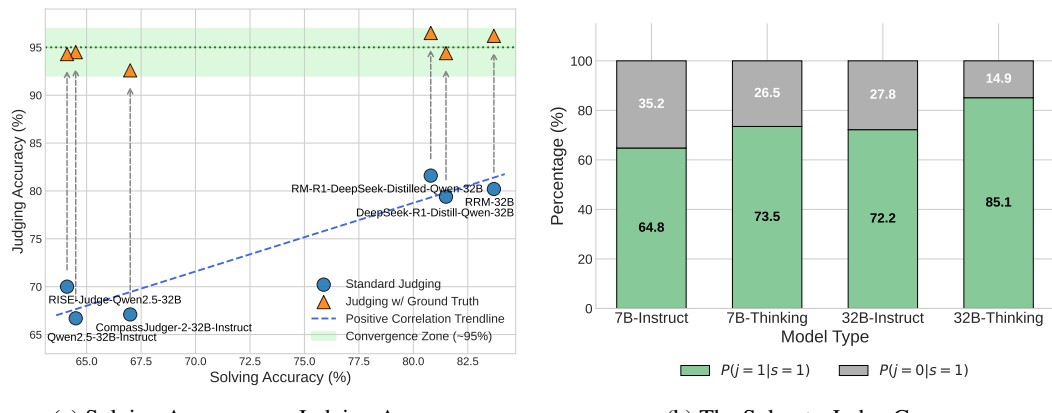

(a) Solving Accuracy vs. Judging Accuracy.    (b) The Solve-to-Judge Gap.

Figure 1: **An empirical analysis of the relationship between solving and judging capabilities.**
**(a)** A positive correlation exists between solving and judging accuracy. Judging accuracy surges
to a convergence zone of 95% when the model is provided with the ground-truth answer. **(b)** A
breakdown of judging performance on problems the model solves correctly. The green portion
represents the accuracy of correct judgments ($P(j = 1|s = 1)$), while the gray portion quantifies
the solve-to-judge gap ($P(j = 0|s = 1)$), revealing that models fail to judge 14.9% to 35.2% of the
problems they are capable of solving. Detailed results are provided in Appendix C.1.

model on preference datasets. Formally, given an input triplet $(x, y_a, y_b, l)$, the generative reward
model $\pi_\theta$ produces a trajectory:

$$\tau = (c, \hat{l}) \sim \pi_\theta(\cdot \mid x, y_a, y_b), \tag{2}$$

where $c$ is the reasoning chain which may include understanding and solving the problem, analysis
of the response, and any other content. $\hat{l} \in \{a, b\}$ is the model's predicted preference label. The
outcome reward is then defined as:

$$R(\tau) = \begin{cases} 1, & \text{if } \hat{l} = l, \\ 0, & \text{otherwise.} \end{cases} \tag{3}$$

This outcome reward is used by RLVR algorithms (such as GRPO (Shao et al., 2024), RLOO (Ah-
madian et al., 2024), and DAPO (Yu et al., 2025b)) to optimize the generative reward model $\pi_\theta$.

### 2.3 PROBLEM MOTIVATION

To explore the relationship between models' problem-solving and judging abilities, we conduct ex-
periments on several popular GRMs across diverse domains using the widely adopted preference
dataset PPE (Frick et al., 2024). We use accuracy (ACC) as the evaluation metric for both problem-
solving and judging abilities here. Our results in Figure 1 reveal a previously overlooked but im-
portant problem. This observation directly motivates our study, which we summarize through the
following findings:

**Judging Ability is Bottlenecked by Problem-Solving Ability.** As shown in Figure 1a, we observe
that the models clearly fall into several clusters. Models with similar problem-solving ability also
exhibit similar judging ability, while models with stronger problem-solving skills achieve higher
judging performance. This indicates that judging ability is fundamentally bottlenecked by problem-
solving ability. A similar finding is also observed in Deepseek-GRM (Liu et al., 2025b), which
uses a large amount of general training data to improve general problem-solving ability, thereby
enhancing judging ability. To further illustrate it, motivated by Zhou et al. (2025), we directly
provide the ground-truth answers in the prompt, i.e., assuming that the model's problem-solving
ability has reached maximum. As shown in Figure 1a, all models achieve a similar level of judging
performance, further demonstrating that problem-solving ability is indeed a bottleneck for judging.

**There is a Significant Solve-to-Judge Gap.** While the above findings suggest that improvements in
problem-solving ability can lead to better judging performance, we observe that even when models

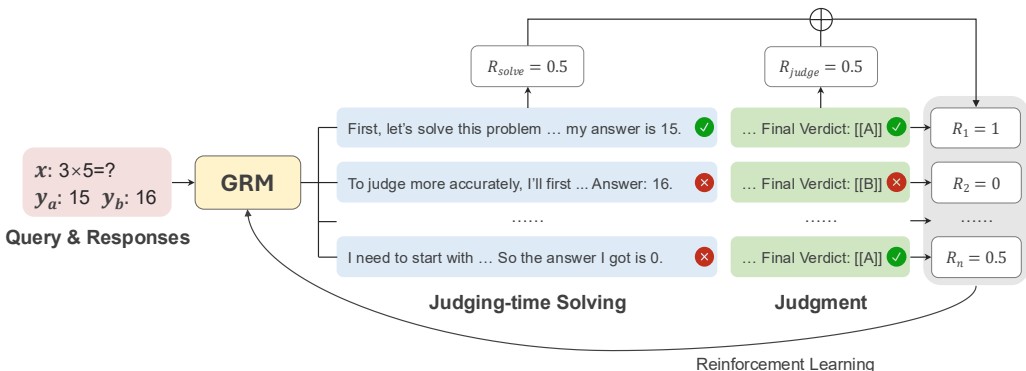

Figure 2: **An overview of our proposed Solve-to-Judge (S2J) method.** Given a query ($x$) and a pair of responses ($y_a, y_b$), the Generative Reward Models (GRMs) is trained to first perform **judging-time solving** (i.e., generate its own solution) before providing a final **judgment**. We derive two distinct reward signals from its output: a solving reward ($\mathcal{R}_{\text{solve}}$) for the correctness of its self-generated solution and a judging reward ($\mathcal{R}_{\text{judge}}$) for the correctness of its final verdict. These rewards are combined and used in a reinforcement learning loop to optimize the GRMs, explicitly training it to leverage its existing solving capabilities to inform and improve its judgments.

are capable of solving certain problems, they often fail to correctly judge them. We term this inconsistency the solve-to-judge gap. To formally analyze this gap, let's denote $s = 1$ as the event where a model solves a problem correctly and $j = 1$ as the event where it judges the provided responses correctly. Ideally, if a model could perfectly leverage its solving capability, its judging accuracy on problems it can solve should be 100%, i.e., $P(j = 1 \mid s = 1) = 1$. However, as shown by the green bars in Figure 1b, this is far from the case. The gap is therefore represented by the failure rate, $P(j = 0 \mid s = 1)$, which is the probability of a model failing to judge correctly a problem it can actually solve. As explicitly quantified by the error rates in Figure 1b, this gap ranges from 14.9% to 35.2%. This finding highlights a key ineffectiveness in existing GRMs: they do not fully utilize their inherent problem-solving ability when performing judging. Bridging this gap is precisely the core problem we aim to address in this paper.

## 3 SOLVE-TO-JUDGE

### 3.1 OVERVIEW

In this section, we describe our solution, Solve-to-Judge (S2J), designed to address the gap discussed in Section 2.3. Figure 2 provides an overview of S2J. In brief, before judging a pair of candidate responses ($y_a, y_b$), S2J first requires the generative reward model to produce its own solution $\hat{y}$ to the user query $x$. S2J then assigns a unified reward that accounts for both the quality of the generated solution and the correctness of the final judgment. Specifically, lower rewards are assigned when the quality of the model's solution and the correctness of its judgment are misaligned, while higher rewards are given when both the solution is of high quality and the judgment is correct. The intuition behind this design is that by granting higher rewards in such aligned cases, S2J encourages the model to link its judgments on its problem-solving process, thereby narrowing the solve-to-judge gap.

### 3.2 METHODOLOGY

**Problem Formulation.** To facilitate the transformation from the model's solving ability into its judging ability, we employ the prompt shown in Figure 3, which instructs the generative reward model $\pi_\theta$ to first solve the user query $x$ itself before judgment. Formally, we redefine Equation 2 as:

$$\tau = (\hat{y}, c, \hat{l}) \sim \pi_\theta(\cdot \mid x, y_a, y_b), \qquad (4)$$

where $\hat{y}$ is the generative reward model's self-generated solution to $x$.

**Reward for judgment.** As in previous work (Chen et al., 2025; Guo et al., 2025b), we use the standard outcome reward to encourage the model to make the correct final judgment, which we define as:

$$\mathcal{R}_{\text{judge}}(\tau) = \begin{cases} 0.5, & \text{if } \hat{l} = l \\ 0, & \text{otherwise.} \end{cases} \tag{5}$$

**Reward for Judging-Time Solving.** Judging-time solving is the key innovation of S2J, as it effectively bridges the *solve-to-judge* gap by requiring the model to first generate its own solution before making a judgment. This internal simulation of the solving process grounds the model's judgment in a deeper and more accurate understanding of the problem, rather than relying on superficial heuristics. Since not all problems can be definitively resolved using a rule-based verifier, we consider two scenarios:

**(1) Objective Tasks.** For tasks with a verifiable ground truth (e.g., mathematics problems with numeric solutions), the reward is computed using a rule-based verifier:

$$\mathcal{R}_{\text{solve}}(\tau) = \begin{cases} 0.5, & \text{if } \textit{Verifier}(\hat{y}, y) = 1, \\ 0, & \text{otherwise,} \end{cases} \tag{6}$$

where *Verifier* is a rule-based function that checks the equivalence between the self-generated solution $\hat{y}$ against the ground truth $y$.

**(2) Subjective Tasks.** For tasks that lack a single definitive ground truth (e.g., creative writing), we employ an auxiliary scalar reward model, denoted as $\text{RM}_{\text{aux}}$. We define $s_l$ as the score of the better response in $\{y_a, y_b\}$, $s_{\neg l}$ as the score of the worse one, and $s_{\hat{y}}$ as the score of the self-generated solution by $\pi_\theta$.

The solving reward is then defined as:

$$\mathcal{R}_{\text{solve}}(\tau) = \begin{cases} 0.5 \cdot \mathbb{1}\left(|s_{\hat{y}} - s_l| < |s_{\hat{y}} - s_{\neg l}|\right), & \text{if } s_l > s_{\neg l}, \\ 0, & \text{otherwise.} \end{cases} \tag{7}$$

The intuition behind this formulation is as follows: (1) The indicator function $\mathbb{1}\left(|s_{\hat{y}} - s_l| < |s_{\hat{y}} - s_{\neg l}|\right)$ determines whether the score of $\hat{y}$ is closer to the better solution than to the worse one. If this condition holds, we regard $\pi_\theta$ as demonstrating strong problem-solving ability for the given query $x$. (2) We compute this reward only when $s_l > s_{\neg l}$, i.e., when $\text{RM}_{\text{aux}}$ is able to correctly handle the partial ordering of a given sample $(x, y_a, y_b)$. If this condition is not met, it indicates that $\text{RM}_{\text{aux}}$ lacks sufficient capability to handle this sample reliably. In such cases, we do not assign a solving reward to the sample, and the overall reward falls back to relying solely on the judgment reward. Note that, since this condition remains consistent across all rollouts sampled from the same training instance $(x, y_a, y_b)$, this design does not introduce reward inconsistencies in RL algorithms such as GRPO and DAPO.

**RLVR with S2J Reward.** Finally, we optimize $\pi_\theta$ using following standard RLVR algorithm (e.g., GRPO, RLOO, or DAPO) with the unified overall reward:

$$\mathcal{R}(\tau) = \mathcal{R}_{\text{solve}}(\tau) + \mathcal{R}_{\text{judge}}(\tau). \tag{8}$$

This composite reward formulation lies at the core of S2J, seamlessly integrating problem-solving and judgment capabilities into a single reinforcement signal, thereby driving consistent improvements in judging accuracy.

## 4 EXPERIMENTS

### 4.1 EXPERIMENTAL SETUP

**Training Data.** Our training data consists of objective tasks (mathematics and knowledge-based question answering) and subjective tasks (human preferences). For mathematics tasks, we randomly sample instances from Math-DPO-10K (Lai et al., 2024). For knowledge-based QA tasks, we synthesize new preference pairs by selecting questions with verifiable answers from WebInstruct-verified (Ma et al., 2025). For each selected query, we prompt Qwen2.5-7B-Instruct and Qwen2.5-32B-Instruct (Qwen et al., 2025) to generate multiple responses, and then apply a rule-based verifier

Table 1: **Performance on four reward model benchmarks.** Our S2J-Qwen2.5-7B model establishes a new state-of-the-art, outperforming all baselines while using significantly less training data. Notably, our method is trained entirely through a self-evolving process without relying on distillation from more powerful models. The best performance is in **bold**. Detailed results for each subset are available in Appendix C.2.

| Model | #Training Pref. Pairs | Self-Evol | PPE Correctness | PPE Preference | Reward Bench | RMB | Average |
|---|---|---|---|---|---|---|---|
| *General LLMs (LLM-as-a-Judge)* | | | | | | | |
| Qwen2.5-7B-Instruct | – | – | 56.7 | 60.4 | 79.1 | 71.4 | 66.9 |
| DeepSeek-R1-Distill-Qwen-7B | – | – | 61.0 | 59.8 | 75.1 | 63.0 | 64.7 |
| *Specialized GRMs* | | | | | | | |
| RISE-Judge-Qwen2.5-7B | 40K | ✗ | 59.1 | 59.8 | 88.2 | 74.7 | 70.5 |
| CompassJudger-2-7B-Instruct | – | ✗ | 58.0 | 64.8 | 84.9 | 76.5 | 71.1 |
| RM-R1$_{Qwen2.5-Instruct-7B}$ | 72.7K | ✗ | 62.0 | 65.2 | 85.2 | 72.9 | 71.3 |
| RM-R1$_{DeepSeek-Distilled-Qwen-7B}$ | 74K | ✓ | 64.7 | 53.5 | 80.1 | 65.5 | 66.0 |
| RRM-7B | 112K | ✓ | 64.6 | 58.8 | 82.2 | 67.9 | 68.4 |
| **S2J-Qwen2.5-7B (Ours)** | **20K** | ✓ | 65.5 | 64.3 | 86.0 | 74.8 | **72.7** |

to check their correctness. The correct responses are taken as positive samples and the incorrect ones as negative samples. Queries without at least one correct and one incorrect response are discarded. For subjective tasks, we sample data from HelpSteer3 (Wang et al., 2025), whose labels are derived from consistent judgments by multiple human annotators, reflecting genuine human preferences. In total, we obtain 20K training samples, which is only 18%–50% used in prior work (Guo et al., 2025b; Chen et al., 2025), yet our approach achieves superior performance.

**Training Implementation.** In all our experiments, we use Qwen2.5-7B-Instruct as the base model. We use Skywork-Reward-V2-Llama-3.1-8B (Liu et al., 2025a) as the auxiliary scalar reward model in Equation 7. We employ DAPO (Yu et al., 2025b) as our RL optimization algorithm, which is an improved variant of GRPO. Our S2J is implemented based on VeRL (Sheng et al., 2025). The training hyperparameters are listed in Table 6.

**Evaluation.** We evaluate our model on four widely-used reward model benchmarks that encompass both objective and subjective tasks: PPE Correctness, PPE Preference (Frick et al., 2024), Reward-Bench (Lambert et al., 2024), and RMB (Zhou et al., 2024) (see details in Appendix D.2). Unless otherwise specified, all inference is conducted using vLLM (Kwon et al., 2023) with a sampling temperature of 1.0 and `top_p` of 1.0. To improve efficiency, for any subset containing more than 512 instances, we randomly sample 512 instances for evaluation. Judgment performance is measured using accuracy (ACC) as the primary metric.

**Baselines.** We compare our S2J-Qwen2.5-7B against two categories of baselines: (1) **General LLMs**: including Qwen2.5-7B-Instruct and DeepSeek-R1-Distill-Qwen-7B (Guo et al., 2025a), evaluated using an LLM-as-a-judge prompt (see Appendix D.1 for prompt details). (2) **Specialized trained GRMs**: including RISE-Judge (Yu et al., 2025a), CompassJudger-2 (Zhang et al., 2025), RM-R1 (Chen et al., 2025), and RRM (Guo et al., 2025b) of comparable size, all of which are trained on extensive preference data and use judgment correctness as the sole reward signal. For fairness, we adopt the original prompts provided in their respective papers; if an original prompt is unavailable, we use the prompt provided in RewardBench.

## 4.2 Main Results

**SOTA performance with Qwen2.5-7B.** Our main results are presented in Table 1. All these methods are built upon the Qwen2.5-7B series models, differing in dataset composition and training methods. Results on the four benchmarks show that our S2J-Qwen2.5-7B model achieves an average score of 72.7% across all benchmarks, establishing a new state-of-the-art performance while using significantly less training data. Moreover, unlike other methods, our S2J achieves this superior performance through a completely self-evolving approach via reinforcement learning, without requiring any distillation from more powerful external models. In summary, our approach delivers high performance with low resource requirements and exceptional efficiency.

Table 2: **Results for the solve-to-judge gap** ($P(j = 0 \mid s = 1)$). We report the relative gap reduction ($\Delta$) with respect to the base model as the primary evaluation metric. Lower values indicate better performance. Our S2J method achieves a **16.2%** reduction in this gap, substantially outperforming all other specialized models and confirming its effectiveness.

| Model | MMLU-Pro | MATH | GPQA | Average | $\Delta$ |
|---|---|---|---|---|---|
| *Base Models* | | | | | |
| Qwen2.5-7B-Instruct | 33.7 | 33.7 | 44.5 | 37.3 | – |
| DeepSeek-R1-Distill-Qwen-7B | 28.1 | 15.3 | 47.2 | 30.2 | – |
| *Specialized GRMs* | | | | | |
| RISE-Judge-Qwen2.5-7B | 36.1 | 25.6 | 43.1 | 34.9 | -2.4 |
| CompassJudger-2-7B-Instruct | 33.3 | 23.5 | 43.0 | 33.3 | -4.0 |
| RM-R1$_{\text{Qwen2.5-Instruct-7B}}$ | 27.1 | 16.9 | 49.7 | 31.2 | -6.1 |
| RM-R1$_{\text{DeepSeek-Distilled-Qwen-7B}}$ | 20.6 | **9.8** | 40.3 | 23.6 | -6.6 |
| RRM-7B | 25.4 | 12.3 | 39.0 | 25.6 | -4.6 |
| **S2J-Qwen2.5-7B (Ours)** | **18.7** | 12.1 | **32.5** | **21.1** | **-16.2** |

**S2J indeed reduces the solve-to-judge gap.** To directly evaluate S2J's ability to bridge the solve-to-judge gap, we employ the proxy metric $P(j = 0 \mid s = 1)$, i.e., the proportion of solved problems that are judged incorrectly, to quantify this gap. A smaller value of $P(j = 0 \mid s = 1)$ indicates a smaller gap. Given that different judge models are built upon different base models, we primarily focus on the reduction values $\Delta$, which denote the decrease in gap relative to each model's respective base model. Table 2 presents our results across three objective benchmarks drawn from subsets of PPE: MMLU-Pro (Wang et al., 2024d), MATH (Hendrycks et al., 2021), and GPQA (Rein et al., 2024). The experimental results demonstrate that our S2J approach achieves significant reduction in the solve-to-judge gap to the baseline model, lowering the average $P(j = 0 \mid s = 1)$ from 37.3% to 21.1%. When compared against other GRMs, our model achieves a remarkable gap reduction of 16.2 percentage points, surpassing the current SOTA by 9.6 percentage points. These findings provide compelling evidence for both the effectiveness and the underlying mechanisms of our S2J reward, as detailed in Section 3.

Based on the judging accuracy improvements and solve-to-judge gap reduction, we further validate our core motivation: explicitly transferring the model's existing solving capabilities into its judging performance through judging-time solving rewards.

## 4.3 ABLATION STUDY

We conduct a series of ablation studies to dissect the key components of our S2J method. We analyze the individual effects of our proposed reward components and the composition of our training data to validate our design choices.

Table 3: **Ablation study on reward components.** The results demonstrate that both components are essential, and their combination synergistically bridges the solve-to-judge gap ($P(j = 0 \mid s = 1)$) and finally improve the judging ability. $\Delta$ is the change relative to the base model.

| | judgment Performance | | Solve-to-Judge Gap | |
|---|---|---|---|---|
| **Reward** | **PPE Correctness** | **PPE Preference** | $P(j = 0 \mid s = 1)$ | $\Delta$ |
| Base Model | 56.7 | 60.4 | 37.3 | - |
| Only $\mathcal{R}_{\text{solve}}$ | 53.4 | 52.5 | 44.5 | +7.2 |
| Only $\mathcal{R}_{\text{judge}}$ | 62.5 | 62.7 | 30.4 | -6.9 |
| **S2J ($\mathcal{R}_{\text{solve}} + \mathcal{R}_{\text{judge}}$)** | **65.5** | **64.3** | **21.1** | **-16.2** |

**Effect of Reward Components.** To investigate the contributions of our reward formulation, we ablate the two key components: the judging-time solving reward $\mathcal{R}_{\text{solve}}$ and the judging outcome reward $\mathcal{R}_{\text{judge}}$. As shown in Table 3, training with **only** $\mathcal{R}_{\text{solve}}$ results in performance degradation across all metrics. This confirms that optimizing for solving accuracy alone is insufficient and

Table 4: **Ablation study on training data composition.** Broadening the data mix improves overall performance. Importantly, incorporating subjective data (Helpsteer3) does not compromise performance on objective tasks, demonstrating the robustness of our approach.

| | judgment Performance | | Solve-to-Judge Gap | |
|---|---|---|---|---|
| **Training Data** | **PPE Correctness** | **PPE Preference** | $P(j = 0 \mid s = 1)$ | $\Delta$ |
| Qwen2.5-7B-Instruct | 56.7 | 60.4 | 37.3 | - |
| + Math-DPO | 62.8 | 54.7 | **20.9** | **-16.4** |
| + WebInstruct | 63.1 | 59.8 | 21.0 | -16.3 |
| **+ Helpsteer3 (Full Mix)** | **65.5** | **64.3** | 21.1 | -16.2 |

can detract from the primary goal of making correct judgments. Training with **only** $\mathcal{R}_{\textbf{judge}}$, which mirrors standard RLVR approaches (Chen et al., 2025; Guo et al., 2025b), yields moderate improvements, decreasing $P(j = 0 \mid s = 1)$ by 6.9%.

The full version of S2J combining both rewards achieves superior performance across all metrics. The synergy of two rewards is most evident in the $P(j = 0 \mid s = 1)$ metric, which decreases by 16.2% over the base model. This represents a remarkable 9.3% absolute decrease compared with using $\mathcal{R}_{\text{judge}}$ alone, empirically validating our core hypothesis: explicitly rewarding the model for correct internal solving during the judging process is crucial for effectively converting its solving ability into reliable judging ability.

**Effect of Training Data Composition.** Here we conduct an ablation study to examine the impact of our training data mixture compositions, as detailed in Table 4. Our experimental results show that: (1) Training S2J solely on objective data (Math-DPO and WebInstruct) can improve the model's performance on PPE Correctness and reduce the solve-to-judge gap on objective tasks. However, training only on such objective tasks leads to a decrease in the model's performance on subjective tasks, i.e. PPE Preference. (2) By introducing subjective preference data from Helpsteer3, our model achieves the best performance on both PPE Correctness and PPE Preference. Crucially, this inclusion of subjective data does not harm the model's performance on objective tasks, and the solve-to-judge gap remains almost unchanged. This result validates that our S2J reward design is effective and robust for both objective and subjective tasks, enabling the model to improve its judging capabilities across diverse domains within a unified training framework.

### 4.4 CASE STUDY

To illustrate the solve-to-judge gap, we present a case study in Table 5. The base model solves the problem correctly but fails to judge the candidate responses, offering a superficial evaluation that leads to an incorrect verdict. This failure to translate its own problem-solving knowledge into evaluation highlights the gap we aim to address. Conversely, our S2J-trained model successfully bridges this gap. By first solving the problem, it establishes a solid foundation for judgment, allowing it to accurately identify the logical fallacy in the incorrect response and validate the correct one. This case vividly shows how S2J converts solving ability into a reliable judging capability.

## 5 RELATED WORK

**Reward Models.** Reward models play a vital role in both the post-training (Ouyang et al., 2022) and automated evaluation (Wang et al., 2023) of LLMs. Traditional reward models are typically trained with the Bradley-Terry objective and output a scalar score indicating the quality of a given response. However, this design often leads to poor calibration and generalization across different prompts and responses (Sun et al., 2025; Zhang et al., 2024). To address these limitations, the generative reward model paradigm is proposed as a more expressive and capable alternative (Zheng et al., 2023).

**Generative Reward Models.** LLM-as-a-Judge (Dubois et al., 2024; Saha et al., 2023) represents an important subset of GRMs that effectively pioneer the use of LLMs' powerful generative abilities to create comprehensive and transparent reasoning processes before making final decisions (Kim et al., 2024; Ankner et al., 2024; Yu et al., 2024; Saha et al., 2025), ultimately resulting in substantially

Table 5: **Case study on a math reasoning task.** Although the base model (Qwen2.5-7B-Instruct) can solve the problem correctly, its judgment process is flawed and superficial (highlighted in red). In contrast, our S2J-trained model grounds its judgment by first solving the problem, enabling it to accurately evaluate the reasoning of both chatbots (highlighted in green). Complete responses for this case study are provided in the Appendix, Table 11.

---

**Question:** Given that the point $(4, 7)$ is on the graph of $y = 3f(2x) + 1$, there is one point that must be on the graph of $y = f(x)$. What is the sum of coordinates of that point?

**Chatbot A (Incorrect):**
... Step 1: ... we get: $f(2 \cdot 4) = \frac{7-1}{3} = 2$. Step 2: ...we can find the point on the graph of $y = f(x)$ by dividing the $x$-coordinate by 2 and the $y$-coordinate by 3, giving us the point $(2, \frac{7-1}{3}) = (2, 2)$. Step 3: The sum of the coordinates of the point $(2, 2)$ is $2 + 2 = \boxed{4}$.

**Chatbot B (Correct):**
... Step 1: ... $7 = 3f(2 \cdot 4) + 1 \implies f(8) = 2$. Step 2: ... Since $f(8) = 2$, the corresponding point on the graph of $y = f(x)$ is $(8, 2)$. Step 3: The sum of the coordinates of this point is: $x + y = 8 + 2 = \boxed{10}$.

**Qwen2.5-7B-Instruct (before S2J): Can solve correctly, but judges incorrectly.**
**Solving Process:** ✓ The model correctly solves the problem on its own, providing a step-by-step derivation: *"...substitute the given point into the equation: $7 = 3f(2 \cdot 4) + 1 \implies f(8) = 2$. Since $f(8) = 2$, the point $(8, 2)$ is on the graph of $y = f(x)$. The sum of the coordinates is $8 + 2 = \boxed{10}$."*

**Judging Process:** ✗ The model's evaluation is contradictory and superficial. *"Evaluation of Chatbot A's Response: ... Properly deduced the corresponding point on the graph of $y = f(x)$ by appropriately adjusting the coordinates. ... Accuracy: The point found is $(2, 2)$. The sum of the coordinates is $2 + 2 = 4$."* (Model praises Chatbot A's steps but describes its incorrect results as accurate.)
*"Evaluation of Chatbot B's Response: ... Incorrectly deduced the corresponding point on the graph of $y = f(x)$. The point $(8, 2)$ is not derived from the given transformation steps. Sum of the coordinates is incorrect as $8 + 2 = 10$."* (Model criticizes Chatbot B for deriving the correct answer.)
**Final Verdict: [[A]]**

**S2J-Qwen2.5-7B (ours): Solve-to-Judge leads to correct judgment.**
**Judging Process:** ✓ *"First, let's solve the client's problem ourselves. We are given that the point $(4, 7)$ is on the graph of $y = 3f(2x) + 1$. This means, substituting $x = 4$, we have: $7 = 3f(2 \cdot 4) + 1 \implies 6 = 3f(8) \implies f(8) = 2$. This tells us that the point $(8, 2)$ is on the graph of $y = f(x)$. So, the sum of the coordinates of this point is $8 + 2 = 10$. So, my answer is: $\boxed{10}$."*
*"Next, let's evaluate the chatbot responses: 1. **Chatbot A's Response**: There was an error in finding the point on the graph of $y = f(x)$. Step 2 incorrectly applies a transformation to the original point $(4, 7)$ instead of using the derived fact that $f(8) = 2$. This led to an incorrect coordinate $(2, 2)$ and the wrong sum. 2. **Chatbot B's Response**: This chatbot correctly solved for $f(8) = 2$ and correctly identified that $(8, 2)$ is the corresponding point on the graph of $y = f(x)$, thus arriving at the correct sum of 10, which is accurate. 3. **Final Verdict**: [[B]]"*

---

enhanced evaluation accuracy. The practical implementation of LLM-as-a-Judge fundamentally relies on sophisticated prompt engineering techniques and carefully designed prompting strategies. Building upon this foundation, researchers have developed specialized training approaches including SFT and DPO to enhance evaluation capabilities (Ye et al., 2024b; Wang et al., 2024b; Mahan et al., 2024; Ye et al., 2024a; Wu et al., 2024; Zhao et al., 2025; Anugraha et al., 2025). The latest developments in LLM reasoning (Guo et al., 2025a; Team et al., 2025; Hu et al., 2025; Xie et al., 2025) have sparked a new wave of research incorporating RLVR techniques into GRM training (Chen et al., 2025; Guo et al., 2025b; Yang et al., 2025; Xu et al., 2025; Yu et al., 2025c). While existing approaches primarily emphasize the accuracy of final verdicts as their reward signal, our research takes a different direction by addressing the solve-to-judge gap. The S2J methodology integrates both judging-time solution rewards and evaluation outcome rewards, specifically designed to facilitate the transfer of inherent problem-solving capabilities into effective evaluation skills.

## 6 CONCLUSION

In this paper, we identified and quantified a significant solve-to-judge gap in generative reward models, revealing that they often fail to correctly evaluate problems they are capable of solving. To bridge this gap, we introduced Solve-to-Judge (S2J), a reinforcement learning method that jointly optimizes for both judgment accuracy and the model's own problem-solving accuracy during the evaluation process. By incorporating a judging-time solving reward, S2J explicitly encourages the model to ground its judgments in its intrinsic problem-solving knowledge. Experiments demonstrate that S2J effectively reduces the solve-to-judge gap by 16.2%, leading to a 5.8% average performance gain across four reward model benchmarks. Crucially, our method achieves state-of-the-art results through a self-evolving training process, without relying on distillation from more powerful models.

ETHICS STATEMENT

Our work adheres to the ICLR Code of Ethics. The research exclusively utilizes publicly available datasets such as Math-DPO, WebInstruct, and HelpSteer, which do not contain personally identifiable or sensitive information. Our data synthesis process involves generating responses from existing large language models and filtering them with a rule-based verifier, without the direct involvement of human subjects. The S2J method itself does not inherently introduce new ethical concerns; rather, by improving the ability to evaluate and align LLMs, it contributes to the broader effort of responsible AI development. We encourage the community to continue investigating and mitigating the potential societal impacts of advanced reward modeling techniques.

REPRODUCIBILITY STATEMENT

To ensure the reproducibility of our results, we've provided comprehensive details throughout the paper and its appendix. The exact composition and sourcing of our training data are described in the Training Data subsection of Section 4.1. Our proposed Solve-to-Judge (S2J) method, including the specific reward formulations, is detailed in Section 3. For implementation, a complete list of training hyperparameters is available in Table 6, along with the prompt template used for training and inference is shown in Figure 3. All evaluation benchmarks are publicly available, and our evaluation setup is specified in the Evaluation subsection of Section 4.1. To facilitate the full replication of our results and to encourage further research, we will release our source code, including scripts for data processing, model training, and evaluation, upon publication of the paper.

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

# A   THE USE OF LARGE LANGUAGE MODELS

In this work, we utilize large language models (LLMs) to assist in checking grammatical errors, performing sentence-level refinement, and verifying formula correctness. All technical contributions, ideas, and claims are proposed by the author without the assistance of LLMs.

# B   S2J IMPLEMENTATION DETAILS

## B.1   PROMPT TEMPLATE

Figure 3 shows the prompt template we use in the Solve-to-Judge (S2J) method. This template is designed to guide the generative reward model $\pi_\theta$ to first independently solve the user query $x$ before evaluating the pair of candidate responses $(y_a, y_b)$.

## B.2   TRAINING HYPERPARAMETERS

We train our S2J-Qwen2.5-7B model using the DAPO algorithm, implemented on the verl framework. The training is conducted on a setup with 8 NVIDIA H800 GPUs. Key hyperparameters for our training process are detailed in Table 6.

# C   FULL EXPERIMENT RESULT

## C.1   DETAILED RESULTS FOR SOLVE-TO-JUDGE GAP ANALYSIS

This section provides the detailed numerical results that support the analyses presented in Figure 1 of the main text. The experiments were conducted on the MMLU-Pro, MATH, and GPQA subsets of the PPE benchmark.

Table 7 corresponds to Figure 1a, detailing the Solving Accuracy (S-Acc), standard Judging Accuracy (J-Acc), and Judging Accuracy with Ground Truth (J-Acc w/ GT). Table 8 corresponds to Figure 1b, providing the precise values for the solve-to-judge gap, quantified as $P(j = 0|s = 1)$.

---

**Prompt Template for S2J**

Please act as an impartial judge and evaluate the quality of the responses provided by two AI Chatbots to the Client's question displayed below.

# For objective tasks.
1. First, you MUST solve the Client's question yourself and put your final answer within \boxed{}. Provide your own solution before proceeding to the evaluation.
# For subjective tasks.
1. First, you MUST solve the Client's question yourself and put your entire solution within <solution> and </solution> tags. Provide your own solution before proceeding to the evaluation.

2. Evaluate the two Chatbot responses based on correctness, referencing your own solution.
3. Output your final verdict by strictly following this format:
'[[A]]' if Chatbot A is better, or '[[B]]' if Chatbot B is better.

[Client Question]
{question}

[The Start of Chatbot A's Response]
{answer_a}
[The End of Chatbot A's Response]

[The Start of Chatbot B's Response]
{answer_b}
[The End of Chatbot B's Response]

---

Figure 3: The prompt template used in our Solve-to-Judge (S2J) method. It instructs the model to first provide its own solution and then judge the given candidate responses.

Table 6: Training hyperparameters for S2J-Qwen2.5-7B.

| Hyperparameter | Value |
|---|---|
| *Data Configuration* | |
| Max prompt length | 4096 |
| Max response length | 8192 |
| *DAPO Algorithm Configuration* | |
| Advantage estimator | GRPO |
| Clip ratio (low) | 0.2 |
| Clip ratio (high) | 0.28 |
| Responses per prompt | 16 |
| Sampling temperature | 1.0 |
| Sampling top-p | 1.0 |
| KL in reward | False |
| KL loss | False |
| *Optimization Configuration* | |
| Optimizer | AdamW |
| Learning rate | 1e-6 |
| Learning rate warmup steps | 10 |
| Weight decay | 0.1 |
| Gradient clipping | 1.0 |
| Batch size | 128 |
| Mini-batch size | 32 |
| Total training steps | 300 |

Table 7: **Detailed Solving and Judging Accuracy on PPE Subsets (MMLU-Pro, MATH, GPQA).** This table provides the data corresponding to Figure 1a. We report Solving Accuracy (S-Acc), standard Judging Accuracy (J-Acc), and Judging Accuracy with Ground Truth (J-Acc w/ GT).

| Model | MMLU-Pro | | | MATH | | | GPQA | | | Average | | |
|---|---|---|---|---|---|---|---|---|---|---|---|---|
| | S-Acc | J-Acc | J-Acc w/ GT | S-Acc | J-Acc | J-Acc w/ GT | S-Acc | J-Acc | J-Acc w/ GT | S-Acc | J-Acc | J-Acc w/ GT |
| *General LLMs* | | | | | | | | | | | | |
| Qwen2.5-7B-Instruct | 52.7 | 62.1 | 78.3 | 66.0 | 66.8 | 94.1 | 32.0 | 51.0 | 75.2 | 50.3 | 60.0 | 82.6 |
| DeepSeek-R1-Distill-Qwen-7B | 61.1 | 63.3 | 84.8 | 88.1 | 83.2 | 90.4 | 59.6 | 46.1 | 73.4 | 69.6 | 64.2 | 82.9 |
| Qwen2.5-32B-Instruct | 72.7 | 70.9 | 95.1 | 74.8 | 73.0 | 95.7 | 46.1 | 56.3 | 92.8 | 64.5 | 66.7 | 94.5 |
| DeepSeek-R1-Distill-Qwen-32B | 81.6 | 78.9 | 94.3 | 91.2 | 93.2 | 96.3 | 71.7 | 66.0 | 92.6 | 81.5 | 79.4 | 94.4 |
| *Specialized GRMs* | | | | | | | | | | | | |
| RISE-Judge-Qwen2.5-7B | 49.8 | 59.0 | 86.1 | 60.2 | 70.5 | 91.6 | 31.3 | 52.0 | 84.4 | 47.1 | 60.5 | 87.4 |
| CompassJudger-2-7B-Instruct | 47.5 | 63.7 | 83.2 | 62.3 | 72.5 | 94.3 | 30.9 | 54.7 | 81.4 | 46.9 | 63.6 | 86.3 |
| RM-R1-DeepSeek-Distilled-Qwen-7B | 59.8 | 67.4 | 90.8 | 89.3 | 89.1 | 94.3 | 61.5 | 51.0 | 80.5 | 70.2 | 69.2 | 88.5 |
| RRM-7B | 55.3 | 66.4 | 92.2 | 92.2 | 87.3 | 95.7 | 60.6 | 54.9 | 86.7 | 69.3 | 69.5 | 91.5 |
| RISE-Judge-Qwen2.5-32B | 70.7 | 72.7 | 94.7 | 77.7 | 79.7 | 96.5 | 43.8 | 57.6 | 91.8 | 64.1 | 70.0 | 94.3 |
| CompassJudger-2-32B-Instruct | 71.9 | 68.4 | 93.4 | 80.5 | 77.9 | 96.1 | 48.6 | 55.1 | 88.3 | 67.0 | 67.1 | 92.6 |
| RM-R1-DeepSeek-Distilled-Qwen-32B | 82.8 | 82.2 | 96.9 | 87.5 | 94.3 | 98.0 | 72.1 | 68.2 | 94.7 | 80.8 | 81.6 | 96.5 |
| RRM-32B | 83.2 | 80.7 | 96.7 | 97.3 | 93.6 | 98.2 | 70.5 | 66.4 | 93.6 | 83.7 | 80.2 | 96.2 |

Table 8: **Detailed Solve-to-Judge Gap** ($P(j = 0|s = 1)$) **on PPE Subsets (MMLU-Pro, MATH, GPQA).** This table provides the data corresponding to Figure 1b. The values represent the percentage of problems that a model judges incorrectly despite being able to solve them correctly.

| Model | MMLU-Pro | MATH | GPQA | Average |
|---|---|---|---|---|
| *General LLMs* | | | | |
| Qwen2.5-7B-Instruct | 33.7 | 33.7 | 44.5 | 37.3 |
| DeepSeek-R1-Distill-Qwen-7B | 28.1 | 15.3 | 47.2 | 30.2 |
| Qwen2.5-32B-Instruct | 24.5 | 25.3 | 37.3 | 29.0 |
| DeepSeek-R1-Distill-Qwen-32B | 16.8 | 6.9 | 24.5 | 16.0 |
| *Specialized GRMs* | | | | |
| RISE-Judge-Qwen2.5-7B | 36.1 | 25.6 | 43.1 | 34.9 |
| CompassJudger-2-7B-Instruct | 33.3 | 23.5 | 43.0 | 33.3 |
| RM-R1-DeepSeek-Distilled-Qwen-7B | 20.6 | 9.8 | 40.3 | 23.6 |
| RRM-7B | 25.4 | 12.3 | 39.0 | 25.6 |
| RISE-Judge-Qwen2.5-32B | 22.7 | 17.1 | 33.0 | 24.3 |
| CompassJudger-2-32B-Instruct | 29.9 | 20.6 | 39.4 | 30.0 |
| RM-R1-DeepSeek-Distilled-Qwen-32B | 12.7 | 4.7 | 24.1 | 13.9 |
| RRM-32B | 14.6 | 5.6 | 24.7 | 14.9 |

## C.2 DETAILED RESULTS ON REWARDBENCH AND RMB

We provide a detailed performance breakdown on the subsets of the RewardBench and RMB benchmarks. Table 9 shows the results on RewardBench, and Table 10 shows the results on RMB.

Table 9: **Detailed results on the subsets of RewardBench.**

| Model | Chat | Chat Hard | Safety | Reasoning | Average |
|---|---|---|---|---|---|
| *General LLMs* | | | | | |
| Qwen2.5-7B-Instruct | 97.5 | 58.3 | 80.4 | 80.0 | 79.1 |
| DeepSeek-R1-Distill-Qwen-7B | 88.0 | 56.3 | 72.1 | 84.0 | 75.1 |
| *Specialized GRMs* | | | | | |
| RISE-Judge-Qwen2.5-7B | 92.2 | 76.5 | 88.0 | 96.1 | 88.2 |
| CompassJudger-2-7B-Instruct | 96.1 | 65.8 | 86.1 | 91.7 | 84.9 |
| RM-R1-Qwen2.5-Instruct-7B | 94.1 | 74.6 | 85.2 | 86.7 | 85.2 |
| RM-R1-DeepSeek-Distilled-Qwen-7B | 88.9 | 66.2 | 78.4 | 87.0 | 80.1 |
| RRM-7B | 87.7 | 70.4 | 80.7 | 90.0 | 82.2 |
| **S2J-Qwen2.5-7B (Ours)** | 97.8 | 74.1 | 84.6 | 87.5 | 86.0 |

Table 10: **Detailed results on the subsets of RMB.**

| Model | Helpfulness | Harmlessness | Average |
|---|---|---|---|
| *General LLMs* | | | |
| Qwen2.5-7B-Instruct | 69.5 | 73.2 | 71.4 |
| DeepSeek-R1-Distill-Qwen-7B | 65.8 | 60.2 | 63.0 |
| *Specialized GRMs* | | | |
| RISE-Judge-Qwen2.5-7B | 72.7 | 76.6 | 74.7 |
| CompassJudger-2-7B-Instruct | 77.5 | 75.5 | 76.5 |
| RM-R1-Qwen2.5-Instruct-7B | 73.0 | 72.9 | 72.9 |
| RM-R1-DeepSeek-Distilled-Qwen-7B | 66.4 | 64.5 | 65.5 |
| RRM-7B | 68.6 | 67.2 | 67.9 |
| **S2J-Qwen2.5-7B (Ours)** | 76.6 | 73.0 | 74.8 |

# D EVALUATION DETAILS

## D.1 PROMPTS FOR BASELINE MODELS

For the LLM-as-a-judge baselines, we use different prompts tailored to the respective models to ensure optimal performance. Figure 4 shows the prompt used for Qwen2.5-7B-Instruct, and Figure 5 shows the prompt used for DeepSeek-R1-Distill-Qwen-7B.

## D.2 BENCHMARK DETAILS

We evaluate our models on four diverse and widely-recognized reward model benchmarks. A brief description of each is provided below.

**PPE (Preference Proxy Evaluations)** The Preference Proxy Evaluations (PPE) (Frick et al., 2024) benchmark is composed of two main proxy task datasets: one focused on large-scale human preferences and another on verifiable correctness.

- **PPE Correctness:** This subset is a verifiable correctness preference dataset created to evaluate a reward model's ability to identify objectively correct LLM-generated responses. It is built using several reputable and verifiable benchmarks, including MMLU-Pro (Wang et al., 2024d), MATH (Hendrycks et al., 2021), GPQA (Rein et al., 2024), MBPP Plus (Liu et al., 2024b), and

**Prompt for Qwen2.5-7B-Instruct**

Please act as an impartial judge and evaluate the quality of the responses provided by two AI assistants to the user question displayed below.

You should choose the assistant that follows the user's instructions and answers the user's question better.
Your evaluation should consider factors such as the helpfulness, relevance, accuracy, depth, creativity, and level of detail of their responses.
Begin your evaluation by comparing the two responses and provide a short explanation.
Avoid any position biases and ensure that the order in which the responses were presented does not influence your decision.
Do not allow the length of the responses to influence your evaluation.
Do not favor certain names of the assistants.
Be as objective as possible.

After providing your explanation, output your final verdict by strictly following this format: "[[A]]" if assistant A is better, "[[B]]" if assistant B is better.

[Client Question]
{question}

[The Start of Chatbot A's Response]
{answer_a}
[The End of Chatbot A's Response]

[The Start of Chatbot B's Response]
{answer_b}
[The End of Chatbot B's Response]

Figure 4: The LLM-as-a-judge prompt used for evaluating the Qwen2.5-7B-Instruct model.

**Prompt for DeepSeek-R1-Distill-Qwen-7B**

Please act as an impartial judge and evaluate the quality of the responses provided by two AI Chatbots to the Client question displayed below.

[Client Question]
{question}

[The Start of Chatbot A's Response]
{answer_a}
[The End of Chatbot A's Response]

[The Start of Chatbot B's Response]
{answer_b}
[The End of Chatbot B's Response]

Output your final verdict at last by strictly following this format: '[[A]]' if Chatbot A is better, or '[[B]]' if Chatbot B is better.

Figure 5: The LLM-as-a-judge prompt used for evaluating the DeepSeek-R1-Distill-Qwen-7B model.

IFEval (Zhou et al., 2023). These cover domains such as general knowledge, mathematics, STEM, coding, and instruction following, respectively.

- **PPE Preference:** This subset is a large-scale human preference dataset designed to measure how well a reward model aligns with direct human judgments. The data is crowdsourced from Chatbot Arena (Chiang et al., 2024) and consists of 16,038 labeled human preference pairs. It is diverse, containing responses from 20 top LLMs and user queries in over 121 languages, including English, Chinese, and Russian. The prompts cover various categories, such as instruction-following, hard prompts, and math problems.

**RewardBench**   RewardBench (Lambert et al., 2024) is a benchmark dataset and codebase created for the evaluation of reward models for language modeling. The benchmark consists of a collection of prompt-chosen-rejected trios, specifically designed to test reward models on comparisons where there are subtle but verifiable reasons (such as factual errors or code bugs) for preferring one response over another. It spans several challenging categories, including general and adversarial chat scenarios (Chat, Chat Hard), safety alignment (Safety), and code and mathematical reasoning (Reasoning).

**RMB (Reward Model Benchmark)**   The RMB (Zhou et al., 2024) is a comprehensive benchmark designed to evaluate how well reward models (RMs) can guide the alignment of large language models with human values like helpfulness and harmlessness. It is constructed from real-world user queries and covers 49 fine-grained scenarios to test the generalization of RMs. The benchmark's results have demonstrated a positive correlation with the downstream alignment performance of RMs.

Table 11: Complete and unabridged responses for the mathematical reasoning case study.

| **Question** |
| --- |
| Given that the point $(4, 7)$ is on the graph of $y = 3f(2x) + 1$, there is one point that must be on the graph of $y = f(x)$. What is the sum of coordinates of that point? |

**Chatbot A's Full Response (Incorrect)**

To solve this problem, we will follow these steps:
1. Substitute the given point $(4, 7)$ into the first equation to find the value of $f(2 \cdot 4)$. 2. Use the value of $f(2 \cdot 4)$ to find the point on the graph of $y = f(x)$. 3. Find the sum of the coordinates of the point on the graph of $y = f(x)$.
Step 1: Substituting the point $(4, 7)$ into the first equation, we get: $7 = 3f(2 \cdot 4) + 1$ Solving for $f(2 \cdot 4)$, we get: $f(2 \cdot 4) = \frac{7-1}{3} = 2$
Step 2: Since the point $(4, 7)$ is on the graph of $y = 3f(2x) + 1$, we can find the point on the graph of $y = f(x)$ by dividing the $x$-coordinate by 2 and the $y$-coordinate by 3, giving us the point $(2, \frac{7-1}{3}) = (2, 2)$.
Step 3: The sum of the coordinates of the point $(2, 2)$ is $2 + 2 = \boxed{4}$.

**Chatbot B's Full Response (Correct)**

To solve this problem, we can follow these steps:
1. Substitute the given point $(4, 7)$ into the equation $y = 3f(2x) + 1$: $7 = 3f(2 \cdot 4) + 1$ $7 = 3f(8) + 1$ $6 = 3f(8)$ $f(8) = 2$
2. Now, we need to find the point on the graph of $y = f(x)$ that corresponds to the point $(4, 7)$ on the graph of $y = 3f(2x) + 1$. Since $f(8) = 2$, the corresponding point on the graph of $y = f(x)$ is $(8, 2)$.
3. The sum of the coordinates of this point is: $x + y = 8 + 2 = \boxed{10}$.

**Qwen2.5-7B-Instruct's (before S2J) Full Judgment (Incorrect)**

Let's evaluate the quality of the two responses to the Client's question.
### Evaluation of Chatbot A's Response 1. **Correctness of Steps**: - Correctly identified and applied the given equation to find $f(8)$. - Properly deduced the corresponding point on the graph of $y = f(x)$ by appropriately adjusting the coordinates. - Sum of the coordinates was correctly calculated. 2. **Accuracy**: - The point found is $(2, 2)$. - The sum of the coordinates is $2 + 2 = 4$.
### Evaluation of Chatbot B's Response 1. **Correctness of Steps**: - Correctly identified and applied the given equation to find $f(8)$. - Incorrectly deduced the corresponding point on the graph of $y = f(x)$. - The point $(8, 2)$ is not derived from the given transformation steps. - Sum of the coordinates is incorrect as $8 + 2 = 10$. 2. **Accuracy**: - The point found is $(8, 2)$. - The sum of the coordinates is $8 + 2 = 10$, which is incorrect based on the transformation logic.
### Conclusion Chatbot A's response is accurate and correctly answers the question. Chatbot B, on the other hand, made a mistake in the derivation of the corresponding point on the graph of $y = f(x)$.
Thus, the final verdict is: [[A]]

**S2J-Qwen2.5-7B's (ours) Full Judgment (Correct)**

First, let's solve the client's problem ourselves. We are given that the point $(4, 7)$ is on the graph of $y = 3f(2x) + 1$. This means, substituting $x = 4$, we have: $7 = 3f(2 \cdot 4) + 1$
Simplifying, we get: $7 = 3f(8) + 1$
Subtracting 1 from both sides, we have: $6 = 3f(8)$
Dividing both sides by 3, we get: $f(8) = 2$
This tells us that the point $(8, 2)$ is on the graph of $y = f(x)$. Hence, the sum of the coordinates of this point is: $8 + 2 = 10$ So, my answer is: $\boxed{10}$
Next, let's evaluate the chatbot responses:
1. **Chatbot A's Response**: There was an error in finding the point on the graph of $y = f(x)$. Instead of the point being $(2, 2)$, it should be $(8, 2)$ based on correctly using $f(8) = 2$ from the original problem transformation. This led to an incorrect coordinate, making the sum wrong.
2. **Chatbot B's Response**: This chatbot correctly solved for $f(8) = 2$ and correctly identified that $(8, 2)$ is on the graph of $y = f(x)$, thus arriving at the correct sum of 10, which is accurate.
3. **Final Verdict**: [[B]]

