# OpenReview forum: "S2J: Bridging the Gap Between Solving and Judging Ability in Generative Reward Models"
_ICLR.cc/2026/Conference — ICLR 2026 Conference Withdrawn Submission_

### Official Review · Reviewer_scZs · 2025-10-30

**Soundness:** 4
**Presentation:** 4
**Contribution:** 3
**Rating:** 6
**Confidence:** 3

**Summary:**

This paper addresses the challenge of improving generative reward models by targeting the gap between judging and solving. The authors empirically observe a gap between a model's ability to solve and judge. The authors propose integrating a traditional judging-based reward with a solving-based reward, thereby aligning evaluation performance with problem-solving ability. The goal is to reduce the solve-to-judge discrepancy while enhancing evaluation through the model’s solving capabilities. Extensive experiments demonstrate that the approach improves judging accuracy, significantly narrows the solve-to-judge gap, and achieves state-of-the-art results with fewer training samples.

**Strengths:**

1. **Clear presentation**: The paper is well-written, with a clearly stated problem and a simple yet effective proposed method. Diagrams and figures are used effectively to support understanding.
2. **Strong empirical motivation**: The authors convincingly demonstrate the existence of the solve-to-judge gap, providing a solid foundation for the necessity of their approach.
3. **Builds on prior work**: The method extends existing approaches with novel observations and solutions that advance the field.
4. **Well-designed experiments**: The experimental results cover multiple benchmarks and highlight the effectiveness of S2J.

**Weaknesses:**

1. **Limited model diversity**: Experiments are conducted only with Qwen models as the base. Including additional model families would strengthen claims of generalizability.
2. **Lack of theoretical discussion**: While the solve-to-judge gap is clearly shown empirically, the paper provides little discussion on why it arises. Is it inherent to LLMs, a consequence of training objectives, or another factor? A deeper analysis would add valuable insight.
3. The main algorithmic contribution is incremental and requires an external expert model for training

**Questions:**

1. What exactly is meant by self-evolving in this context?
2. The paper claims the method does not require powerful external models for distillation, yet in the subjective setting, an external model is used to provide scores. Does this imply that a judge is still required to train the new S2J judge? If so, how can both statements be true a the same time?
3. Is there any discussion or hypothesis on the root cause of the solve-to-judge gap? Could it be addressed earlier during pretraining or supervised fine-tuning?


**Questions that did not impact the rating**
- For subjective tasks, your method relies on an auxiliary scalar model. Could the framework be extended into a two-step or iterative scheme, where earlier versions of the learned model serve as the auxiliary model?
- In Table 1, the best scores are not highlighted in bold. Is there a reason for this choice?
- In Figure 1, when ground truth is provided, error is ~5%. When models solve the task, the error rises to 15–35%. Does the ground truth include explanations? If not, why is the error higher when the model correctly solves the task?
- Regarding the S2J prompt (Figure 3), how was this prompt designed? Was there any ablation to measure how prompt choice affects S2J effectiveness?

---

> ### Author Response · Authors · 2025-11-25
> **Response to Reviewer scZs (1/2)**
>
> We sincerely thank you for your positive assessment and for highlighting our "strong empirical motivation" and "well-designed experiments."
>
> Below, we address your comments on model diversity, theoretical insights, and clarifications regarding our methodology.
>
> > **W1: Limited model diversity.**
>
> To demonstrate the generalizability of S2J, we extended our experiments to **Llama-3.1-8B-Instruct**. Despite previous findings [1] suggesting that Llama models can be more challenging to optimize via RL compared to Qwen, S2J achieved substantial gains:
>
> | Model | #Pairs | PPE Corr. | PPE Pref. | RewardBench | RMB | Avg. | $P(j=0 \mid s=1)$ |
> | :--- | :---: | :---: | :---: | :---: | :---: | :---: | :---: |
> | Llama-3.1-8B-Instruct | - | 37.4 | 45.7 | 70.3 | 50.5 | 51.0 | 62.5 |
> | Qwen2.5-7B-Instruct | - | 56.7 | 60.4 | 79.1 | 71.4 | 66.9 | 37.3 |
> | **S2J-Llama-3.1-8B-Instruct** | 20K | 60.1 | 63.5 | 79.3 | 71.3 | 68.5 | 35.1 |
> | **S2J-Qwen2.5-7B-Instruct** | 20K | 65.5 | 64.3 | 86.0 | 74.8 | 72.7 | 21.1 |
>
> S2J reduced the solve-to-judge gap by **27.4%** and improved average judgment accuracy by **17.5%**, confirming that our method is effective across different model families.
>
> > **W2 & Q3: Theoretical discussion and root cause of the gap.**
>
> We appreciate the push for a deeper analysis. Based on qualitative analysis of error cases, we hypothesize that the gap arises from "Robustness to Distraction" during the transition from solving to judging.
>
> The **Distraction Hypothesis** suggests that when solving independently, the model only relies on its internal knowledge. However, during judgment, the model is exposed to candidate responses, which often contain distractors (e.g., persuasive but incorrect reasoning, logical fallacies). We observe that in gap cases ($s=1, j=0$), the model often gets misled by a plausible-sounding incorrect response in the prompt, causing it to abandon its own correct internal solution.
>
> As shown in Figure 1, larger models (32B) and Thinking models have a naturally smaller gap, suggesting that resistance to such contextual interference is a capability that scales with model size and reasoning depth. S2J explicitly forces the model to prioritize its own generated solution (via the Judging-time Solving Reward) before attending to the candidates, thereby increasing its robustness against these distractors.
>
> > **W3, Q1 & Q2: Clarification on Self-evolving and External Models.**
>
> We apologize for any confusion and would like to clarify our terminology and setting.
>
> 1.  **Definition of Self-evolving (Q1):** In our context, self-evolving means the model improves its judging capability from scratch via Reinforcement Learning (RL), without needing Supervised Fine-Tuning (SFT) on trajectories distilled from a stronger teacher model. The model explores and optimizes its own reasoning paths.
>
> 2.  **Distinction from Distillation (W3, Q2):** SFT Distillation typically involves cloning the token-level trajectories of a superior model, which requires a powerful teacher to generate full responses. In contrast, our approach (RLVR) uses a scalar RM merely as an **Outcome Verifier**. For subjective tasks where rule-based verification is impossible, the scalar RM provides a sparse, numerical signal indicating if the solution is acceptable.
>
> Crucially, this proxy is conditional: it only takes effect when the scalar RM assigns a higher score to the ground-truth good response than to the bad one. Otherwise, the solving reward $\mathcal{R}\_{\text{solve}}$ follows the judgment outcome $\mathcal{R}\_{\text{judge}}$. Consequently, the primary supervision signal in our framework remains derived from the preference labels in the training dataset.

---

> ### Author Response · Authors · 2025-11-25
> **Response to Reviewer scZs (2/2)**
>
> > **Questions that did not impact the rating**
>
> * **Iterative Scheme:** Yes, this is a promising direction! We will explore reducing dependency on external models for subjective tasks in RLVR in our future research.
>
> * **Bold formatting in Table 1:** Thank you for pointing this out. We will correct this in the final revision to highlight the best results in bold and the second-best underlined.
>
> * **Figure 1 (Ground Truth vs. Solved):** With Correct Solving, even if the model solves it correctly internally, it must still actively defend its solution against the incorrect response's logic. As mentioned in the "Distraction Hypothesis" above, models often fail this defense step and succumb to the incorrect response's persuasiveness.
>
> * **S2J Prompt Design:** The prompt is adapted from [2]. We tested S2J-Qwen2.5-7B on the PPE benchmark using the minimal judge prompt (Fig 5). We found that the model still spontaneously solves the problem before judging, yielding comparable performance. The difference is negligible (65.5% vs. 64.9%), indicating that the solve-to-judge capability has been effectively internalized.
>
> | Prompt Type | PPE Corr. | PPE Pref. |
> | :--- | :---: | :---: |
> | Original Prompt (Ours) | 65.5 | 64.3 |
> | Minimal Prompt (RM-R1) | 64.9 | 64.7 |
>
> ---
>
> We hope our response addresses your concerns. If you have any further questions, we are more than willing to provide additional clarifications.
>
> [1] Cognitive Behaviors that Enable Self-Improving Reasoners, or, Four Habits of Highly Effective STaRs
> [2] RM-R1: Reward Modeling as Reasoning

---

> > ### Comment · Reviewer_scZs · 2025-11-27
> >
> > I appreciate the authors' detailed replies and the extended experiment results. Since all of my concerns have been addressed, I have raised my score accordingly.

---

> ### Comment · Reviewer_scZs · 2025-11-27
>
> Due to some technical issue, I am not able to edit my original review to update my rating.
>
> After carefully reading the response from the authors, I would like to update my rating to "Accept (8)".

---

> > ### Author Response · Authors · 2025-11-28
> > **Thank You for Your Recognition and Support**
> >
> > Dear Reviewer scZs,
> >
> > We sincerely thank you for your active engagement and for taking the time to read our response. We are delighted to hear that our clarifications and the additional experiments on Llama-3.1 have addressed your concerns.
> >
> > We have noted your comment regarding the technical issue with updating the rating system. **We sincerely appreciate your decision to raise the score to "Accept (8)".** Your constructive feedback has significantly helped us strengthen the paper, particularly regarding model diversity and the analysis of the gap's root causes.
> >
> > Thank you again for your time and support.
> >
> > Best regards, The Authors

---

### Official Review · Reviewer_opy4 · 2025-10-30

**Soundness:** 2
**Presentation:** 2
**Contribution:** 2
**Rating:** 4
**Confidence:** 4

**Summary:**

Generative reward models (GRMs) are widely used with large language models (LLMs) for evaluation, but they often show a solve-to-judge gap—failing to judge 14–37% of queries they can solve. The proposed Solve-to-Judge (S2J) method links a GRM’s solving and judging abilities during training, reducing this gap by 16.2% and improving judgment accuracy by 5.8%. S2J achieves state-of-the-art results with a smaller dataset and without relying on external model distillation.

**Strengths:**

1. The authors reveal the solve-to-judge gap through extensive experiments, showing that current GRMs incorrectly evaluate 14–37% of problems they can solve, thereby establishing a comprehensive understanding of this limitation.

2. The authors introduce Solve-to-Judge (S2J), a method that jointly optimizes solving and judging capabilities, effectively narrowing the solve-to-judge gap and enhancing the judgment performance of GRMs.

3. The authors demonstrate that S2J reduces the solve-to-judge gap by 16.2% and improves judgment performance by 5.8% across multiple benchmarks.

**Weaknesses:**

1. I am concerned about the existence and validity of the “solve-to-judge gap,” as the experimental design does not sufficiently rule out other possible factors that might explain the observed phenomenon.
2. There is a lack of clear explanation of the link between LLMs’ problem-solving abilities and their judgment capabilities, and a stronger theoretical or empirical justification for this claim is needed.
3. The authors’ claim is unclear, and the experimental design requires a more detailed and transparent explanation.

**Questions:**

See the weaknesses.

---

> ### Author Response · Authors · 2025-11-25
> **Response to Reviewer opy4**
>
> We sincerely thank you for your thoughtful review. We appreciate that you recognized our work's contribution in identifying the solve-to-judge gap and the effectiveness of our S2J method in narrowing it.
>
> Below, we address your concerns regarding the validity of the gap and the relationship between solving and judging capabilities with further clarifications and additional empirical evidence.
>
> > **W1 & W3: Validity of the solve-to-judge gap and experimental design transparency.**
>
> We understand your concern regarding whether other factors might explain the observed phenomenon. To clarify the existence and validity of the solve-to-judge gap, we performed a deeper breakdown analysis comparing Independent Solving vs. Judging-time Solving.
>
> **1. Defining the Gap via Context Shift:**
> The core of the solve-to-judge gap is a degradation in the model's solving capability when it shifts from a pure solving context to a judging context.
>
> * **Independent Solving:** When the model is simply asked to "solve this problem," it often succeeds.
> * **Judging-time Solving:** When the model is asked to "judge these responses," and implicitly needs to solve the problem to verify correctness, it often gets distracted by the incorrect candidate responses (distractors) or fails to activate its solving knowledge, leading to incorrect judgments.
>
> **2. Statistical Evidence:**
> We conducted a specific analysis on the PPE-MATH dataset using the Qwen2.5-7B-Instruct model to quantify this degradation:
>
> * **The Phenomenon:** We identified a subset of problems where the model could solve the problem correctly in isolation. However, when placed in the judging prompt (without S2J training), the model failed to generate the correct solution internally during the judgment process in **52.1%** of these cases. This massive drop confirms that the Judging Context significantly impairs the model's ability to access its own solving knowledge—this is precisely the "gap" we define.
> * **S2J's Effectiveness:** After applying our S2J training, this failure rate (solving correctly alone but failing during judgment) dropped significantly from **52.1% to 13.2%**.
> * **Link to Judgment:** In Section 2.3, we established that correct judging-time solving is a prerequisite for correct judgment. By fixing the solving degradation (bridging the gap), S2J directly recovers judgment performance.
>
> We hope this breakdown clarifies that the gap is a tangible failure of knowledge transfer due to context shift.
>
>
>
> > **W2: Link between problem-solving and judgment capabilities.**
>
> To clarify the link between problem-solving and judgment capabilities, we provide the following empirical justifications. Our work builds on the finding that judging ability is bottlenecked by problem-solving ability (Line 151). As shown in Figure 1a, we observe a strong positive correlation between a model's solving accuracy and its judging accuracy across varying model sizes and families. Crucially, our experiment in Figure 1a shows that when we explicitly provide the Ground Truth (GT) in the prompt (simulating perfect solving ability), the judging accuracy for all models surges to a convergence zone of ~95%. This empirically proves that the primary factor limiting judgment performance is indeed the lack of a correct internal solution.
>
> Furthermore, our conclusion is consistent with the findings in [1]. They explored whether verifying a problem’s solution is easier than solving it and found that "for a fixed model, the judge’s accuracy closely mirrors that of the solver." This supports the causal link that a judge cannot accurately verify a complex reasoning step if it cannot derive the correct step itself. S2J works by explicitly forcing the model to demonstrate this prerequisite solving capability before rendering a verdict, thereby removing the bottleneck.
>
> ---
>
> We hope our response addresses your concerns. If you have any further questions, we are more than willing to provide additional clarifications.
>
> [1] JudgeBench: A Benchmark for Evaluating LLM-based Judges

---

> ### Author Response · Authors · 2025-11-28
> **Looking Forward to Your Reply**
>
> Dear Reviewer opy4,
>
> We thank you again for your thoughtful review and for recognizing the importance of the problem we aim to solve.
>
> We wanted to gently follow up to ensure our rebuttal has sufficiently clarified your concerns regarding the validity of the **solve-to-judge gap** and the link between problem-solving and judging capabilities.
>
> In our response, we provided:
>
> 1. A breakdown analysis of **Independent Solving vs. Judging-time Solving**, highlighting how the "Judging Context" significantly degrades solving performance (the distraction hypothesis).
> 2. Statistical evidence showing S2J effectively recovers this performance drop.
> 3. References to recent work (e.g., JudgeBench) that support the causal link between solving and judging.
>
> We hope that these explanations and the additional evidence have clarified the mechanisms behind the gap and the effectiveness of our proposed method. We remain available to answer any further questions you might have before the discussion period ends.
>
> Best regards, The Authors

---

### Official Review · Reviewer_Bp3L · 2025-11-01

**Soundness:** 2
**Presentation:** 2
**Contribution:** 2
**Rating:** 2
**Confidence:** 4

**Summary:**

This paper identifies a "solve-to-judge gap" in generative reward models (GRMs), where models fail to correctly judge queries they can solve, and proposes Solve-to-Judge (S2J), a RLVR method that jointly optimizes solving and judging on preference data. Using Qwen2.5-7B-Instruct, S2J reduces the gap by 16.2% on average.

**Strengths:**

Clearly defines and quantifies the solve-to-judge gap problem, providing empirical evidence across multiple datasets.

**Weaknesses:**

1. The training approach for subjective tasks could benefit from further exploration; at present, it resembles distillation from one RM to another, though I recognize that the core objective here is indeed to train an RM from scratch.

2. It would be valuable to include experiments assessing generalization to other model architectures and larger models (e.g., beyond 14B parameters). For a methods-focused paper, evaluating a single model may limit the demonstration of broader applicability.

3. The reward design appears somewhat heuristic; incorporating sensitivity analyses (e.g., examining the method's robustness to variations in reward formulation) could strengthen this aspect.

4. A comparison with other reward models, such as scalar reward models (e.g., Bradley-Terry reward models) [1], would be a helpful addition.

5. The work could be further enriched with additional analytical experiments, for instance:

   - Evaluating the probability that S2J correctly judges responses when the solving step is incorrect;

   - Investigating whether S2J's gains stem from longer chains of thought rather than the mechanism itself, perhaps by conducting experiments with constraints on generation token limits and their impact on judgment accuracy.



[1] Rethinking Bradley-Terry Models in Preference-Based Reward Modeling: Foundations, Theory, and Alternatives

**Questions:**

Please refer to Weaknesses.

---

> ### Author Response · Authors · 2025-11-25
> **Response to Reviewer Bp3L (1/2)**
>
> We sincerely thank you for your time and constructive feedback. We appreciate that you recognized the definition and quantification of the "solve-to-judge gap" as a clear strength of our work.
>
> We have conducted additional experiments to address your concerns regarding generalization (adding Llama-3.1-8B), comparisons with scalar RMs, and further analytical investigations.
>
> > **W1: Training approach for subjective tasks resembles distillation.**
>
> In the RLVR setting, rule-based verification is challenging for subjective queries (e.g., creative writing). To address this, we employ a scalar RM as a proxy verifier for the "Judging-time Solving" phase to bridge the gap. Crucially, this proxy is conditional: it only takes effect when the scalar RM assigns a higher score to the ground-truth "good" response than to the "bad" one. Otherwise, the solving reward $\mathcal{R}\_{\text{solve}}$ follows the judgment outcome $\mathcal{R}\_{\text{judge}}$. Consequently, the primary supervision signal in our framework remains derived from the preference labels in the training dataset. We will explore reducing dependency on external models for subjective tasks in RLVR in our future research.
>
> > **W2: Generalization to other model architectures.**
>
> To demonstrate the broad applicability of S2J beyond the Qwen family, we applied our method to **Llama-3.1-8B-Instruct**.
>
> Recent studies [1] have noted that Llama models can be more challenging to optimize via RL compared to Qwen. However, as shown in the table below, S2J significantly boosts Llama's performance.
>
> | Model | #Pairs | PPE Corr. | PPE Pref. | RewardBench | RMB | Avg. | $P(j=0 \mid s=1)$ |
> | :--- | :---: | :---: | :---: | :---: | :---: | :---: | :---: |
> | Llama-3.1-8B-Instruct | - | 37.4 | 45.7 | 70.3 | 50.5 | 51.0 | 62.5 |
> | Qwen2.5-7B-Instruct | - | 56.7 | 60.4 | 79.1 | 71.4 | 66.9 | 37.3 |
> | **S2J-Llama-3.1-8B-Instruct** | 20K | 60.1 | 63.5 | 79.3 | 71.3 | 68.5 | 35.1 |
> | **S2J-Qwen2.5-7B-Instruct** | 20K | 65.5 | 64.3 | 86.0 | 74.8 | 72.7 | 21.1 |
>
> **Key Findings:**
>
> * **Performance:** S2J improved the average judgment accuracy by **+17.5%** (from 51.0% to 68.5%).
> * **Gap Reduction:** Crucially, the solve-to-judge gap was drastically reduced by **27.4%** (from 62.5% to 35.1%), further validating that our method works across different architectures.
>
> > **W4: Comparison with scalar reward models.**
>
> We have added a comparison with enhanced baseline Scalar Reward Models (RMs). The data below is sourced from the respective benchmark papers (blank cells indicate unavailable corresponding data).
>
> | Model | PPE Corr. | PPE Pref. | RewardBench | RMB | Average |
> | :--- | :---: | :---: | :---: | :---: | :---: |
> | ArmoRM-Llama3-8B-v0.1 | 61.0 | 60.6 | 89.0 | 72.5 | 70.8 |
> | InternLM2-7B-Reward | 60.0 | 62.1 | - | 76.6 | - |
> | Skywork-Reward-Llama-3.1-8B | 61.0 | 62.4 | - | **77.0** | - |
> | Eurus-RM-7b | 62.0 | - | 81.6 | 75.6 | - |
> | **S2J-Qwen2.5-7B-Instruct (Ours)** | **65.5** | **64.3** | 86.0 | 74.8 | **72.7** |
>
> S2J outperforms these strong scalar baselines with the highest average score. Furthermore, unlike scalar models that output a single opaque score, S2J is a Generative Reward Model that provides interpretable reasoning and solutions, offering greater transparency for downstream alignment.

---

> ### Author Response · Authors · 2025-11-25
> **Response to Reviewer Bp3L (2/2)**
>
> > **W5: Additional Analytical Experiments.**
>
> **(1) Judgment accuracy when solving is incorrect ($P(j=1 \mid s=0)$).**
>
> * Base Model (Qwen2.5-7B-Instruct): 55.4%
> * S2J-Qwen2.5-7B-Instruct: 58.6%
>
> Since the task is pairwise classification (random chance = 50%), these values are close to random guessing. This indicates that our performance gains do not stem from the model "getting lucky" when it fails to solve the problem. Instead, the gains come primarily from increasing the instances where the model correctly solves the problem and then successfully transfers that knowledge to the judgment (i.e., reducing the $P(j=0 \mid s=1)$ gap).
>
> **(2) Impact of generation length (Chain of Thought).**
>
> To determine if gains come solely from longer reasoning, we compared average output token lengths on the PPE Correctness benchmark:
>
> | Model | Avg. Acc. | Avg. Length (Tokens) |
> | :--- | :---: | :---: |
> | Qwen2.5-7B-Instruct | 66.9 | 136.7 |
> | DeepSeek-R1-Distill-Qwen-7B | 64.7 | 2225.8 |
> | RISE-Judge-Qwen2.5-7B | 70.5 | 519.5 |
> | CompassJudger-2-7B-Instruct | 71.1 | 335.6 |
> | RM-R1-DeepSeek-Distilled-Qwen-7B | 66.0 | 2086.4 |
> | RRM-7B | 68.4 | 999.5 |
> | **S2J-Qwen2.5-7B (Ours)** | **72.7** | **550.2** |
>
> S2J achieves SOTA performance with a moderate length (~550 tokens), comparable to other efficient judges. Crucially, it significantly outperforms "Thinking" models (like DeepSeek-R1-Distill) that generate >1000 tokens but suffer from a larger solve-to-judge gap (as shown in Table 2). This confirms that S2J's effectiveness stems from the mechanism (bridging solving and judging), not merely from increased computation or length.
>
> ---
>
> We hope our response addresses your concerns. If you have any further questions, we are more than willing to provide additional clarifications.
>
> [1] Cognitive Behaviors that Enable Self-Improving Reasoners, or, Four Habits of Highly Effective STaRs

---

> ### Author Response · Authors · 2025-11-28
> **Looking Forward to Your Reply**
>
> Dear Reviewer Bp3L,
>
> Thank you again for your time and the constructive feedback provided in your initial review.
>
> We are writing to kindly follow up on our previous response. In our rebuttal, we have endeavored to address your primary concerns by:
>
> 1. **Generalization:** Conducting additional experiments on **Llama-3.1-8B-Instruct**, where S2J reduced the solve-to-judge gap by 27.4% and improved judgment accuracy by 17.5%.
> 2. **Comparisons:** Adding a detailed comparison with strong **Scalar Reward Model baselines** (e.g., ArmoRM, Skywork-Reward), demonstrating that S2J achieves superior performance.
> 3. **Analysis:** Providing further analysis on generation length and judging accuracy when solving fails ($P(j=1|s=0)$).
>
> We hope these new results satisfactorily address your concerns regarding the method's robustness and applicability. As the discussion period draws to a close, we would greatly appreciate it if you could let us know if there are any remaining questions we can clarify.
>
> Best regards, The Authors

---

### Note · Authors · 2026-01-05

I have read and agree with the venue's withdrawal policy on behalf of myself and my co-authors.